# DYRK1A Negatively Regulates CDK5-SOX2 Pathway and Self-Renewal of Glioblastoma Stem Cells

**DOI:** 10.3390/ijms22084011

**Published:** 2021-04-13

**Authors:** Brianna Chen, Dylan McCuaig-Walton, Sean Tan, Andrew P. Montgomery, Bryan W. Day, Michael Kassiou, Lenka Munoz, Ariadna Recasens

**Affiliations:** 1Charles Perkins Centre and School of Medical Sciences, Faculty of Medicine and Health, The University of Sydney, Sydney, NSW 2006, Australia; bche4229@uni.sydney.edu.au (B.C.); dmcc3573@alumni.sydney.edu.au (D.M.-W.); stan6439@sydney.edu.au (S.T.); 2School of Chemistry, Faculty of Science, The University of Sydney, Sydney, NSW 2006, Australia; andrew.montgomery@sydney.edu.au (A.P.M.); michael.kassiou@sydney.edu.au (M.K.); 3QIMR Berghofer Medical Research Institute, 300 Herston Road, Herston, QLD 4006, Australia; bryan.day@qimrberghofer.edu.au

**Keywords:** glioblastoma, cancer stem cells, DYRK1A, CDK5, SOX2, bone morphogenetic protein 4 (BMP4)

## Abstract

Glioblastoma display vast cellular heterogeneity, with glioblastoma stem cells (GSCs) at the apex. The critical role of GSCs in tumour growth and resistance to therapy highlights the need to delineate mechanisms that control stemness and differentiation potential of GSC. Dual-specificity tyrosine phosphorylation-regulated kinase 1A (DYRK1A) regulates neural progenitor cell differentiation, but its role in cancer stem cell differentiation is largely unknown. Herein, we demonstrate that DYRK1A kinase is crucial for the differentiation commitment of glioblastoma stem cells. DYRK1A inhibition insulates the self-renewing population of GSCs from potent differentiation-inducing signals. Mechanistically, we show that DYRK1A promotes differentiation and limits stemness acquisition via deactivation of CDK5, an unconventional kinase recently described as an oncogene. DYRK1A-dependent inactivation of CDK5 results in decreased expression of the stemness gene SOX2 and promotes the commitment of GSC to differentiate. Our investigations of the novel DYRK1A-CDK5-SOX2 pathway provide further insights into the mechanisms underlying glioblastoma stem cell maintenance.

## 1. Introduction

Glioblastoma is the most common primary brain tumour. Current standard of care includes surgical resection, radiotherapy and temozolomide-based chemotherapy. The median survival of 15 months for glioblastoma patients has not changed over the past decades and glioblastoma remains an incurable disease with a dismal prognosis [1]. A major contributor to the failure of therapeutic modalities is aberrant transcriptional and cellular hierarchy within each glioblastoma, resulting in complex inter- and intra-tumoral heterogeneity. At the apex of this hierarchy sits a small population of glioblastoma stem cells (GSCs).

GSCs demonstrate two principal features of stem cells: self-renewal and differentiation; and are able to move in either direction along the hierarchy to become more or less differentiated [2,3]. GSCs initiate tumours, promote tumour growth and are crucial drivers of therapeutic resistance. The resistance of GSCs is enabled by the spectrum of multipotent microstates of highest entropy that GSCs occupy, which creates numerous escape routes when exposed to cytotoxic therapies. On the other hand, as cell differentiate, they are constrained to a limited escape program and are proposed to be more sensitive to therapy [4].

Several markers, including CD133, CD15, L1CAM and SOX2 are enriched in GSCs, although none of the markers exclusively and comprehensively mark GSCs [3]. Of these, SOX2 is a well-established stem cell master regulator overexpressed in glioblastoma, particularly in the undifferentiated GSC populations [5,6]. High levels of SOX2 have been associated with glioblastoma aggressiveness and worse prognosis [7,8]. Downregulation or upregulation of SOX2 through RNA interference demonstrated that SOX2 possesses an important role in the maintenance and self-renewal capacity of GSCs [9]. Furthermore, SOX2 is one the transcription factors, together with POU3F2, OLIG2 and SALL2, sufficient to reprogram differentiated glioma cells into induced GSCs [10]. Nevertheless, while the critical role of SOX2 in the maintenance of GSC is now clearly described, we have limited understanding of the upstream mechanisms regulating SOX2 expression.

Dual-specificity tyrosine phosphorylation-regulated kinase 1A (DYRK1A) plays important roles in the developmental and adult neurogenesis [11]. The distinguishing feature of DYRK1A is its ability to autophosphorylate a tyrosine residue in the activation loop during protein translation, which is necessary for protein folding and achieving full catalytic activity. The mature DYRK1A protein is constitutively active and DYRK1A kinase activity depends on its expression levels, a phenomenon known as dosage effect. The active DYRK1A expressed at sufficient levels then phosphorylates serine and threonine residues in numerous nuclear and cytoplasmic proteins [11].

DYRK1A expression in neural stem cells is necessary for cessation of the proliferative division and entry into differentiation pathways [12]. DYRK1A regulates exit of neural stem cells from the cell cycle through mechanisms involving cyclin D1 degradation and p27 stabilisation [13,14]. In addition to the anti-proliferative activity, DYRK1A promotes differentiation of neuronal progenitor cells by suppressing canonical stem cell signalling pathway Notch [12,15]. Thus, while physiological levels of DYRK1A are critical for brain development, deregulation of DYRK1A expression is associated with developmental and cognitive deficits in Down Syndrome, Autism Spectrum Disorder, Alzheimer’s and Parkinson’s Disease [11].

In cancer, aberrant DYRK1A expression and activity plays a critical role in the progression of multiple tumours [11]. In glioblastoma, DYRK1A causes cell cycle arrest via phosphorylating the DREAM complex component LIN52 [16]. Our recent study has demonstrated that DYRK1A promotes cyclin B degradation in GSCs, leading to decreased CDK1 activity and cell cycle arrest [17]. Together, these studies demonstrate that DYRK1A promotes cessation of GSC proliferation, a crucial step for differentiation to occur. However, whether DYRK1A also regulates differentiation pathways in GSC, as it does neural stem cells [12,15], remains unclear. 

In this study, we examine the role of DYRK1A in glioblastoma stem cell regulation. We show that DYRK1A inhibition results in a block to glioblastoma stem cells differentiation and that DYRK1A inhibition insulates the self-renewing population of GSCs from potent differentiation-inducing signals. Mechanistically, we delineate a novel DYRK1A-CDK5-SOX2 pathway underlying self-renewal potential of glioblastoma stem cells and show that DYRK1A activity is necessary for the differentiation of glioblastoma stem cells.

## 2. Results

### 2.1. GSC Undergo Diverse Differentiation Pathways in Response to BMP4

GSC lines cultured in serum-free condition, where media is supplemented with growth factors EGF and FGF, maintain their de-differentiated stem-like phenotype [18]. Replacement of growth factors with the bone-morphogenetic protein 4 (BMP4) induces pronounced astrocyte differentiation in some, but not all GSC [19]. We have developed a glioblastoma patient-derived cell line resource Q-Cell in which glioblastoma cells are maintained as glioma stem cell cultures [20]. Full genomic and proteomic characterisation, clinical data and subtype assignment is publicly available from the Q-Cell webpage (https://www.qimrberghofer.edu.au/q-cell/, accessed on 7 April 2021).

Given the variable differentiation response of GSC to BMP4 [19], we first assessed how MMK1 and HW1 cells, two genetically distinct glioblastoma stem cell lines from our Q-Cell panel, respond to differentiation cues. MMK1 cells carry PTENF56V, IDH1Y183C mutations and heterozygous CDKN2A/B deletions. HW1 cells carry EGFRA289V mutation and heterozygous CDKN2A/B deletions [20]. Dose-dependent and time-course experiments revealed that replacing growth factors (GF) with 20 ng/mL BMP4 for seven days induced pronounced morphological changes in MMK1 cells, suggesting differentiation (Figure 1A,B). When treated with this dose of BMP4, both MMK1 and HW1 cell lines stopped proliferating, as evidenced by reduced cell counts and percentage of Ki67-positive cells (Figure 1C,D). However, consistent with previous reports [19], we observed variability in the degree and type of BMP4-induced differentiation. In the MMK1 cell line, BMP4 treatment significantly increased glial fibrillary acidic protein (GFAP) levels (Figure 1E), suggesting astrocyte differentiation. Nevertheless, immunofluorescence imaging revealed that while a subpopulation of cells exhibited strong upregulation of the astrocyte marker GFAP, some cells failed to efficiently upregulate GFAP (Figure 1F). Based on βIII-tubulin immunofluorescence images, all MMK1 cells changed morphology to bigger, rounder and more dendritic cells, resulting in overall increase of βIII-tubulin levels (Figure 1E,F). Similarly, all HW1 cells showed increased βIII-tubulin expression and changed morphology to dendritic, rounder cells when GF were replaced by BMP4. However, HW1 did not upregulate the astrocyte marker GFAP (Figure 1G,H) when treated with BMP4, suggesting differentiation into a more neuronal lineage.

As analysing levels of astrocyte marker GFAP did not provide uniform and quantifiable measurement of the differentiation commitment, we next assessed levels of the stemness/self-renewal marker SOX2. Replacement of the growth factors EGF/FGF with BMP4 resulted in significantly reduced SOX2 mRNA (Figure 1I) and protein (Figure 1J) expression in both MMK1 and HW1 cell lines. BMP4-induced differentiation was further confirmed by reduced expression of the stemness marker nestin (Figure 1K,L).

In summary, while BMP4 had a cytostatic effect and induced differentiation in tested GSC lines, the differentiation lineage differs not only between cell lines, but also between cells within the same cell line. Thus, monitoring decline in the stemness marker SOX2 offers a more reliable quantification of the differentiation commitment and has been used in our further experiments.

### 2.2. DYRK1A Limits Self-Renewal Capacity of GSC

To investigate whether DYRK1A is necessary for the differentiation commitment of GSCs, DYRK1A was depleted in MMK1 and HW1 cells using DYRK1A-targeting siRNA (Figure 2A). Proliferation, SOX2 expression and morphology were assessed in DYRK1A-depleted cells cultured in the self-renewal medium (i.e., EGF/FGF) or treated with the differentiation-inducing agent BMP4. When cultured in the self-renewal medium (GF), DYRK1A depletion increased proliferation of MMK1 cells (Figure 2B). Replacement of GF with BMP4 reduced proliferation, in line with results in Figure 1C, and the anti-proliferative effect of BMP4 was rescued by DYRK1A depletion (Figure 2B). In the HW1 cell line, DYRK1A depleted cells remained proliferative (Figure 2C), which is in agreement with our recent study reporting that DYRK1A inhibition increases proliferation of RB-deficient MMK1 cells but does not change proliferation of RB-proficient HW1 cells, since RB and DYRK1A are functionally redundant [17]. Nevertheless, HW1 cells cultured in BMP4-containing media stopped proliferating and this effect was prevented by DYRK1A knockdown (Figure 2C).

We next assessed SOX2 expression in DYRK1A-depleted cells. When cells were cultured in the self-renewal medium containing EGF/FGF, DYRK1A knockdown increased SOX2 levels in both MMK1 and HW1 cell lines (Figure 2D,E, red bars). In line with Figure 1I,J, replacement of EGF/FGF with BMP4 significantly reduced SOX2 expression (Figure 2D,E, green bars). Importantly, this effect was less prominent in DYRK1A-depleted cells (Figure 2D,E, blue bars). Finally, immunofluorescence imaging of βIII-tubulin revealed that, in the self-renewal medium, DYRK1A-depleted MMK1 and HW1 cells were smaller and rounder compared to scramble siRNA-transfected cells. We also observed that the morphological changes induced by BMP4 and marking differentiation, i.e., enlarged and dendritic morphology, were absent in DYRK1A-depleted cells (Figure 2F,G). Taken together, these results suggest that DYRK1A limits the self-renewal ability of glioblastoma stem cells, and DYRK1A expression/activity is necessary for differentiation commitment.

### 2.3. DYRK1A Regulates CDK5 Pathway in Glioblastoma Cells

Our data revealed that DYRK1A is a negative regulator of the stemness marker SOX2 (Figure 2). In order to delineate how DYRK1A downregulates SOX2 expression, we analysed our proteomic and phosphoproteomic data obtained from U251 glioblastoma cells infected with doxycycline (DOX)-inducible DYRK1A shRNA (sh-DYRK1A U251) or treated with DYRK1A inhibitor ALGERNON [17]. We observed that while SOX2 is not a direct DYRK1A substrate, SOX2 protein levels increased by 1.46-fold when DYRK1A was inhibited with DYRK1A inhibitor ALGERNON (Figure 3A). Analysis of hyperphosphorylated sites revealed enrichment of cyclin-dependent kinase 5 (CDK5) substrates [17] and we found that 51% of the reported CDK5 substrates (www.phosphosite.org) were hyperphosphorylated after genetic (i.e., DOX-induced DYRK1A shRNA) and pharmacological (i.e., employing ALGERNON) DYRK1A inhibition (Figure 3B). Together, these data suggest that DYRK1A inhibition activates the CDK5 pathway. Importantly, CDK5 has been reported to phosphorylate and thereby activate the transcription factor CREB, leading to increased transcription of CREB-target gene SOX2 [21]. We therefore hypothesised that DYRK1A inhibition leads to increased SOX2 expression via activation of CDK5.

CDK5 is an unconventional CDK whose activity is regulated by the non-cyclin proteins p35 and p39 [22]. The binding of these coactivators to CDK5 is necessary and sufficient to fully activate CDK5, as it tethers the conformational changes of the CDK5 activation loop to an active state. The activity of CDK5 is further enhanced by the phosphorylation of tyrosine residue 15 (Y15) by a variety of tyrosine kinases [22].

To validate the impact of DYRK1A inhibition on CDK5 pathway activity, we used the same genetic and pharmacological approach to inhibit DYRK1A as in our proteomic study [17], i.e., DOX-inducible DYRK1A knockdown and ALGERNON-mediated inhibition of DYRK1A activity in serum-grown differentiated U251 glioblastoma cells. Both genetic (72 h) and pharmacological DYRK1A inhibition increased the expression of CDK5-coactivators p35 and p39 (Figure 3C,D). DYRK1A knockdown decreased phosphorylation of CDK5 and increased the expression of total CDK5 (Figure 3E), whereas inhibition of DYRK1A activity with ALGERNON induced CDK5 phosphorylation without affecting total CDK5 levels (Figure 3F). Given the inconsistent changes to CDK5 phosphorylation upon genetic and pharmacological DYRK1A inhibition, we concluded that DYRK1A inhibition likely leads to CDK5 activation by increasing p35 and p39 levels.

Furthermore, DYRK1A inhibition induced phosphorylation of CREB without changing the total levels of this transcription factor (Figure 3G,H). We also found that DYRK1A knockdown or treatment of U251 cells with DYRK1A inhibitor increased SOX2 expression (Figure 3I,J). These results suggest that DYRK1A inhibition activates CDK5 by increasing the protein levels of its coactivators p35 and p39. This is associated with CREB activation and increased expression of the stem cell marker SOX2. In summary, DYRK1A appears to act as a negative regulator of the CDK5-CREB-SOX2 axis.

We next investigated the DYRK1A-CDK5-CREB-SOX2 pathway in glioblastoma stem cells. HW1 cells were transfected with siRNA targeting DYRK1A or treated with DYRK1A inhibitor ALGERNON. As observed in the differentiated U251 cells (Figure 3C,D), DYRK1A inhibition in HW1 cells resulted in increased expression of the coactivator p35 (Figure 4A,B); ALGERNON also increased p39 levels. Moreover, we found that DYRK1A targeting in GSC moderately increased the expression of total CDK5 protein but had a marked effect on tyrosine Y15 phosphorylation of CDK5 (Figure 4C,D), all of which is associated with CDK5 activation. In further support, DYRK1A inhibition increased phosphorylation of CDK5-target CREB (Figure 4E,F). DYRK1A knockdown increased SOX2 levels in a time-dependent manner (Figure 4G), but this effect was not observed after ALG treatment (Figure 4H). These results indicate that DYRK1A negatively regulates CDK5 activity and limits the expression of SOX2 in GSC.

### 2.4. DYRK1A Regulates SOX2 Expression via CDK5

To validate the DYRK1A-CDK5-SOX2 axis in glioblastoma cells, we analysed how DYRK1A inhibition changes SOX2 expression in CDK5-depleted cells. If CDK5 is a downstream target of DYRK1A and links DYRK1A inhibition to SOX2 accumulation, the absence of CDK5 would prevent or halt DYRK1A inhibition-induced effects. To test this hypothesis, U251 cells were transfected with CDK5 siRNA one day prior to DYRK1A inhibition (DOX or ALGERNON). Immunoblotting confirmed CDK5 and DYRK1A knockdown (Figure 5A,B). As previously shown in Figure 3I,J, genetic and pharmacological DYRK1A inhibition differently affected total CDK5 levels. Importantly, however, both genetic and pharmacological DYRK1A inhibition consistently increased the expression of SOX2 (Figure 5A,B). In contrast, DYRK1A inhibition failed to increase SOX2 expression in CDK5-depleted cells (blue bar, Figure 5A,B), suggesting that CDK5 is necessary for DYRK1A regulation of SOX2.

To further validate these findings in GSC, HW1 cells were transfected with CDK5 and DYRK1A siRNA alone and in combination (Figure 5C). Immunoblots demonstrated that DYRK1A knockdown increased SOX2 levels (red bar, Figure 5C). In line with data obtained with the differentiated U251 cells, CDK5 knockdown blocked increases in SOX2 expression after DYRK1A inhibition (blue bar, Figure 5C). Furthermore, DYRK1A mRNA expression negatively correlated (*p* < 0.005) with CDK5 mRNA expression in the TCGA glioblastoma dataset (Figure 5D), which is in line with mechanistic data presented here. Together, these findings confirm that DYRK1A negatively regulates SOX2 expression via CDK5 pathway and thus is necessary for the differentiation commitment of glioblastoma stem cells.

## 3. Discussion

Glioblastoma stem cells have been functionally defined as cells possessing tumour-propagating and self-renewal potentials. Other features of GSCs include their ability to differentiate into multiple cellular lineages, expression of defined markers and low frequency in a tumour sample [3,23]. The pool of GSC in tumours is maintained by their asymmetric division and also by the ability of differentiated glioblastoma cells to de-differentiate [10,24]. The contribution of GSC to tumour propagation and therapeutic resistance is supported by studies showing that triggering GSC differentiation reduces their tumour-propagating potential and increases response to therapy [25]. Thus, targeting these tumour-initiating and -propagating cells appears crucial to improve the survival rates of glioblastoma patients and requires detailed understanding of key glioblastoma stemness regulators, such as SOX2. SOX2 expression limits differentiation commitment [19] and drives a highly proliferative, growth factor-responsive self-renewal state of GSC [26]. On the other side, knockout of SOX2 attenuates the fitness and self-renewal of GSC [27] and impairs glioblastoma invasiveness [28] and tumorigenicity [9].

In this study, we demonstrate that DYRK1A kinase is a negative regulator of SOX2 and that DYRK1A is necessary for the differentiation commitment of GSCs. While DYRK1A and SOX2 have previously been linked together in neuronal progenitors [29,30], mechanistic understanding of the DYRK1A-SOX2 axis is lacking. We discovered that the atypical CDK5 kinase is the key signalling molecule in the DYRK1A-SOX2 network. Mechanistically, we show that DYRK1A inhibition activates the CDK5 pathway by increasing the expression of CDK5 co-activators p35 and p39. Activation of CDK5 leads to phosphorylation of CREB and activation of its target gene SOX2. These findings suggest that DYRK1A attenuates the CDK5-CREB-SOX2 pathway by maintaining low levels of p35 and/or p39. This is most likely through the ability of DYRK1A to mark various proteins for proteosomal degradation [31,32,33] and aligns with previous studies reporting DYRK1A-regulated degradation of cyclin B and cyclin D [13,14,17].

Although the role of CDK5 was initially thought to be restricted to neuronal development [34], a growing amount of evidence demonstrates that CDK5 also plays a vital role in cancer. CDK5, p35 and p39 have been found over-expressed in various tumours and high CDK5 levels correlate with poor prognosis in myeloma, breast, lung and colorectal cancers [21]. In glioblastoma, CDK5 over-expression segregates within mesenchymal patient clusters, indicating dependence of non-mesenchymal tumours on CDK5 activity [35,36]. At the cellular level, CDK5 regulates cell proliferation, apoptosis, angiogenesis, inflammation and immune response [37]. Few recent studies convincingly demonstrate oncogenic functions for CDK5 in glioblastoma as well. Specifically, CDK5 phosphorylates TRIM59 and activates the STAT3 pathway [38], controls the expression of the oncogene PD-L1 [39] and regulates mitochondrial dynamics via DRP1 phosphorylation [40]. In GSC, CDK5 causes SOX2 accumulation and maintains their self-renewal potential [21]. The oncogenic roles of CDK5 are further supported by the anti-proliferative efficacy of CDK5 inhibitors [35,36]. In addition, CDK5 inhibition improved efficacy of the standard-of-care therapeutics in glioblastoma, pancreatic and breast metastasis models [35,36]. Whether this synergistic effect is due to promoting differentiation of cancer stem cells, which is believed to improve therapy outcome [41], remains to be determined. Collectively, these studies imply CDK5 as a therapeutic target in GSC. Importantly, however, our discovery that DYRK1A attenuates CDK5 activity is important for the translation of DYRK1A inhibitors into glioblastoma therapy. DYRK1A inhibitors might have a limited value in efforts to target glioblastoma, as DYRK1A inhibition would most likely support the self-renewal programs in GSC.

In summary, this study demonstrates that DYRK1A plays a substantial role in promoting GSC differentiation, in addition to its role in the cessation of the cell cycle in glioblastoma cells [16]. We uncovered that DYRK1A weakens the CDK5-SOX2 network in GSC, which is crucial for the activation of self-renewal programs. This is an important finding as it provides further insight into the maintenance and survival of GSC. Additionally, this study supports the notion that DYRK1A is a tumour suppressor kinase, at least in glioblastoma.

## 4. Materials and Methods

### 4.1. Cell Culture

Glioblastoma cell line U251 (Cat# 09063001) was purchased from the European Collection of Authenticated Cell Cultures (ECACC, Salisbury, UK) in 2014 and authenticated by Cell Bank Australia in 2020 using short tandem profiling. Generation of DOX-inducible DYRK1A U251 cells (sh-DYRK1A U251) has been previously described [17]. Cells were cultured in DMEM supplemented with 10% FBS (InterPath, Heidelberg West, VIC, Australia) and Antibiotic-Antimycotic solution (Life Technologies, Carlsbad, CA, USA) at 37 °C and 5% CO_2_. Patient-derived HW1 and MMK1 glioblastoma stem cell lines were obtained from the publicly available Q-Cell panel of primary brain cancer cell lines established at the QIMR Berghofer Medical Research Institute (https://www.qimrberghofer.edu.au/commercial-collaborations/partner-with-us/q-cell/, accessed on 7 April 2021; also described in ref [20]). The protocols were approved by the Human Ethics Committee of the Royal Brisbane & Women’s Hospital (RBWH 2004/161). HW1 and MMK1 were cultured in KnockOut DMEM/F-12 supplemented with Stem Pro SFM, 2 mM GlutaMAX-ICTS, 20 ng/mL EGF, 10 ng/mL FGF-β and Antibiotic-Antimycotic solution (all Life Technologies) as adherent cells on flasks coated with MatriGel (Corning, NY, USA). The protocols were approved by the Human Ethics Committee of the Royal Brisbane & Women’s Hospital (RBWH 2004/161). Cell cultures were routinely tested for mycoplasma infection, and culturing did not exceed 15 passages.

### 4.2. Transfections

Cells (70–80% confluence) were transfected with 5 nM negative control (Cat# 4390843), *DYRK1A* (Cat# 4390824) and *CDK5* (Cat# 439082) siRNA and Lipofectamine RNAiMAX (all from Life Technologies) according to manufacturer’s instructions.

### 4.3. Antibodies and Reagents

Antibodies against CREB (Cat# 9197), GAPDH (Cat# 97166S), GFAP (Cat# 12389), Ki67 (Cat# 9449) p35 (Cat# 2680S), p39 (Cat# 3275S), pCREB (Cat# 9198), SOX2 (Cat# 3579), mouse IgG HRP-linked (Cat# 7076) and rabbit IgG HRP-linked (Cat# 7074) were purchased from Cell Signalling Technology (Danvers, Massachusetts, USA). Alexa488-conjugated anti-mouse IgG (Cat# A11012) and Alex594-conjugated anti-rabbit IgG (Cat# A11012) were purchased from Life Technologies. The antibody against nestin (Cat# MAB1259) was purchased from R&D Signalling Technology (Minneapolis, Minnesota, USA). Antibodies against CDK5 (Cat# sc-6247) and DYRK1A (Cat# sc-100376) were purchased from Santa Cruz Biotechnology (Dallas, TX, USA). Antibodies against p-CDK5 (Y15) (SAB450427) and βIII-tubulin (Cat# T8860), BMP4 (Cat# SRP3016) and Doxycycline (Cat# D9891) were purchased from Sigma Aldrich (Saint Louis, MO, USA). ALGERNON was synthesised and characterised in-house as previously reported [42].

### 4.4. Western Blotting

Protein concentrations were determined with Pierce BCA assay kit (ThermoFisher Scientific, Waltham, MA, USA), following manufacturer’s instructions. In total, 20–40 μg of protein were resolved (2 h, 95 V) on 4–12% SDS-PAGE gels and transferred onto PVDF membranes using iBlot 2, P3 for 7 min (all Life Technologies). Blocking with 5% skim milk (RT, 1 h) was followed by overnight incubation at 4 °C with primary antibodies in 5% BSA in TBS-T. Membranes were incubated with secondary antibodies in 1% skim milk in TBS-T (RT, 1 h). Detection was performed with Immobilon Western HRP Substrate Luminol-Peroxidase reagent (Merck Millipore, Darmstadt, Germany) and the ChemiDoc MP Imaging System (Bio-Rad, Hercules, CA, USA). Densitometry quantification was done with ImageLab software (Bio-Rad, Hercules, CA, USA).

### 4.5. Immunofluorescence

Cells were fixed with ice-cold 4% PFA (20 min, RT), blocked in 5% normal goat serum/PBS (30 min) and incubated with antibodies against βIII-tubulin (1:1000), GFAP (1:100), nestin (1:100), SOX2 (1:100) and Ki67 (1:400). Secondary antibodies were Alexa488-conjugated anti-mouse IgG and Alexa594-conjugated anti-rabbit IgG (1:500, Life Technologies). Cell nuclei were counterstained using Prolong Gold mounting media with DAPI (Life Technologies). Images were acquired with a Zeiss Axio Scope.A1 microscope using ZEN 2–blue edition software (Lite version, Zeiss, Jena, Germany). Images were processed using Fiji.

### 4.6. Nuclear Staining

Cells were stained with NUCLEAR-ID Red DNA stain (30 min, 37 °C, 5% CO_2_), washed and mounted with Dako Fluorescence Mounting Medium (ENZO, Farmingdale, NY, USA). Images were acquired on a Zeiss upright fluorescence Axio Scope.A1 microscope and analysed using ImageJ. In each replicate, at least 3 randomly chosen images were taken and number of nuclei quantified after setting the threshold automatically, using Li filter and “Analyse particle” function in Batch mode.

### 4.7. RT-PCR

Total RNA from cells was extracted using RNeasy mini kit (Qiagen, Hilden, Germany), retrotranscribed and amplified using Applied Biosystems High-Capacity cDNA Reverse Transcription kit (Life Technologies) as per manufacturer’s instructions. RT-PCR was performed using KAPA SYBR FAST Universal 2X qPCR Master Mix and Qiagen QuantiTect Primer Assays using standard procedures in a LightCycler 480. Threshold cycles (Ct) were calculated using the LightCycler^®^ 480 software (all Roche, Basel, Switzerland). Relative quantification using the comparative Ct method was used to analyse the data output. Values were expressed as fold change over corresponding values for the control by the 2^−∆∆*C*t^ method. Primers were from QIAGEN: GAPDH (Cat# QT00079247), GFAP (Cat# QT00081151), nestin (Cat# QT00235781), SOX2 (Cat# QT00209685), TUBB3 (Cat# QT00083713).

### 4.8. Statistical Analysis

All data are expressed as the mean ± standard error of the mean (SEM). Statistical comparisons were performed with GraphPad Prism software (v7, GraphPad Software Inc, San Diego, CA, USA), as indicated in each figure legend. In all analyses, the null hypothesis was rejected at the 0.05 level. No statistical methods were used to pre-determine sample size but our sample sizes are equivalent to those reported in previous similar publications.

## Figures and Tables

**Figure 1 ijms-22-04011-f001:**
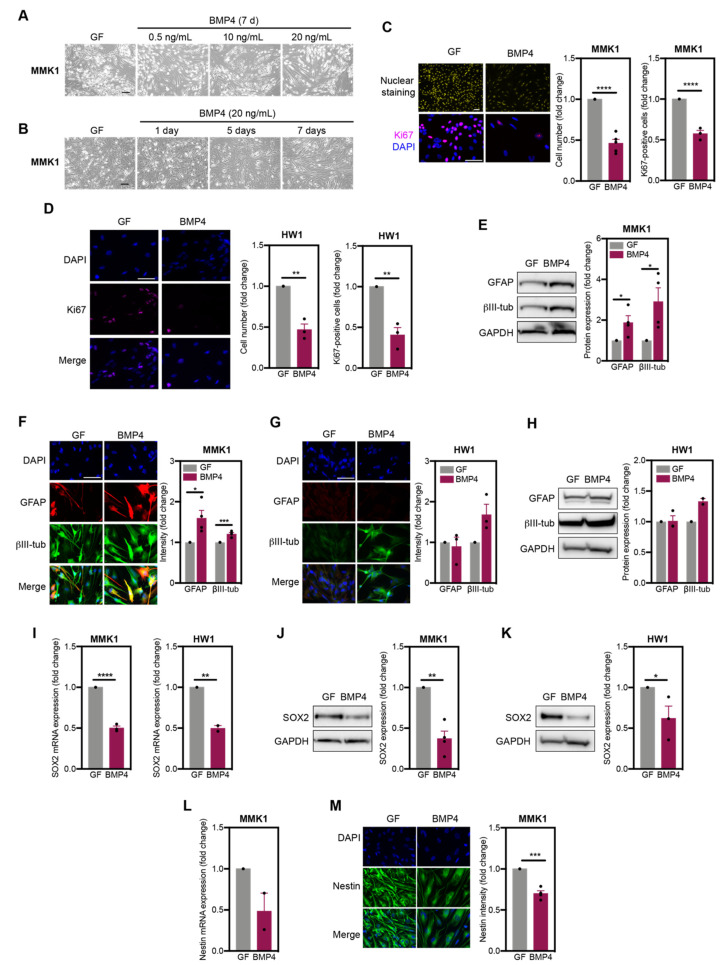
BMP4 induces differentiation in glioblastoma stem cells. (**A**,**B**) Microscopy images of MMK1 cells cultured with growth factors (GF) or bone-morphogenetic protein 4 (BMP4). Scale bar: 10 μm. For all remaining figure panels, MMK1 and HW1 cells were cultured with GF to maintain their stem-like phenotype. To induce differentiation, GF were replaced with BMP4 (20 ng/mL, 7 days). (**C**,**D**) Nuclear staining and Ki67 immunofluorescence images with quantifications of MMK1 (**C**) and HW1 (**D**) cells cultured with GF or BMP4. DAPI was used to visualise cell nuclei (blue). Scale bar: 10 μm. (*n* = 3–5). (**E**,**F**) Immunofluorescence (**E**) and immunoblot (**F**) images with quantification of GFAP and βIII-tubulin (βIII-tub) in MMK1 cells cultured with GF or BMP4 (*n* = 3–4). Scale bar: 10 μm. (**G**,**H**) Immunofluorescence (**G**) and immunoblot (**H**) images and quantification of GFAP and βIII-tubulin (βIII-tub) in HW1 cells cultured with GF or BMP4 (*n* = 3–4). Scale bar: 10 μm. (**I**) RT-PCR analysis of SOX2 mRNA in MMK1 and HW1 cells cultured with GF or BMP4 (*n* = 2). (**J**,**K**) Immunoblot images and quantification of SOX2 expression in MMK1 (**J**) and HW1 (**K**) cells cultured with GF or BMP4 (*n* = 3–4). (**L**) RT-PCR analysis of nestin mRNA in MMK1 cells cultured with GF or BMP4 (*n* = 2). (**M**) Immunofluorescence images and quantification of nestin in MMK1 cells cultured with GF or BMP4 (*n* = 4). All bar graphs represent mean ± SEM (two-tailed, unpaired *t*-test, * *p* < 0.05, ** *p* < 0.01, *** *p* < 0.001, **** *p* < 0.0001).

**Figure 2 ijms-22-04011-f002:**
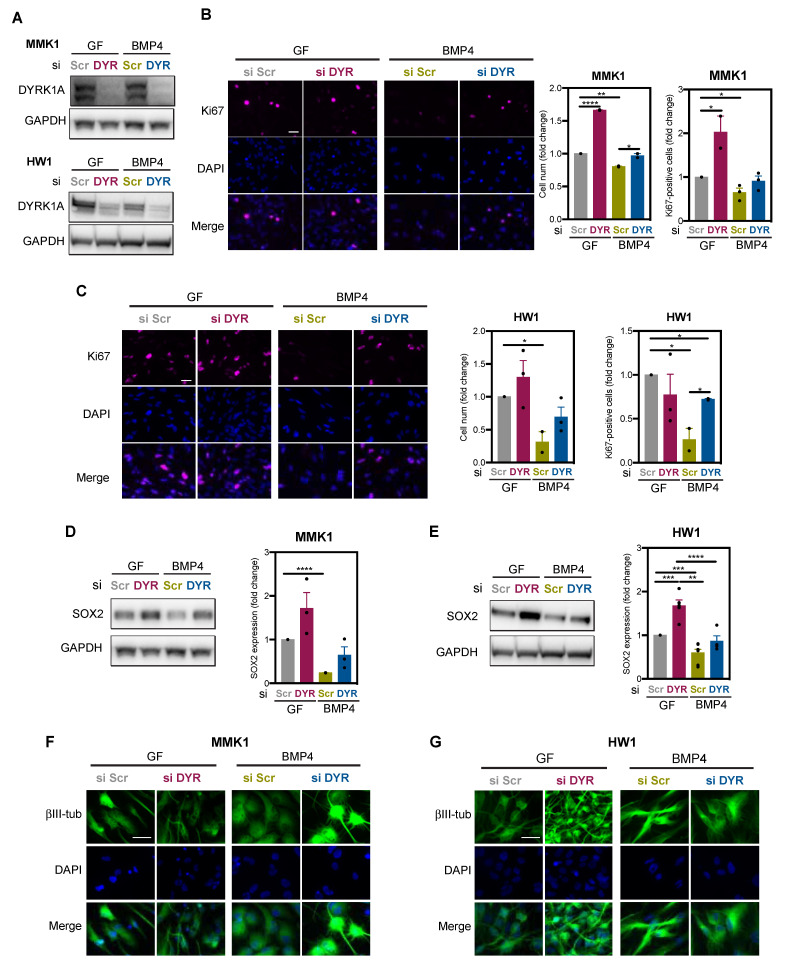
DYRK1A is essential for glioblastoma stem cells differentiation. MMK1 and HW1 cells were cultured with GF to maintain their stem-like phenotype. To induce differentiation, GF were replaced with BMP4 (20 ng/mL, 7 days). (**A**) Immunoblot images of DYRK1A expression in MMK1 and HW1 cells transfected with scramble (si Scr) and DYRK1A (si DYR) targeting siRNA, cultured with GF or BMP4 (7 days). (**B**,**C**) Immunofluorescence images and quantification of Ki67-positive MMK1 (**B**) and HW1 (**C**) cells transfected with scramble (si Scr) and DYRK1A (si DYR) targeting siRNA, cultured with GF or BMP4 (7 days). DAPI was used to visualise cell nuclei. Scale bar: 10 μm. (*n* = 2–3). (**D**,**E**) Immunoblot images and quantification of SOX2 expression in MMK1 (**D**) and HW1 (**E**) cells transfected with scramble (si Scr) and DYRK1A (si DYR) targeting siRNA, cultured with GF or BMP4 (7 days). (*n* = 2–4). (**F**,**G**) Immunofluorescence images and quantification of βIII-tubulin (βIII-tub) in MMK1 (**F**) and HW1 (**G**) cells transfected with scramble (si Scr) and DYRK1A (si DYR) targeting siRNA, cultured with GF or BMP4 (7 days). Scale bar: 10 μm. All bar graphs represent mean ± SEM (two-tailed, unpaired t-test, * *p* < 0.05, ** *p* < 0.01, *** *p* < 0.001, **** *p* < 0.0001).

**Figure 3 ijms-22-04011-f003:**
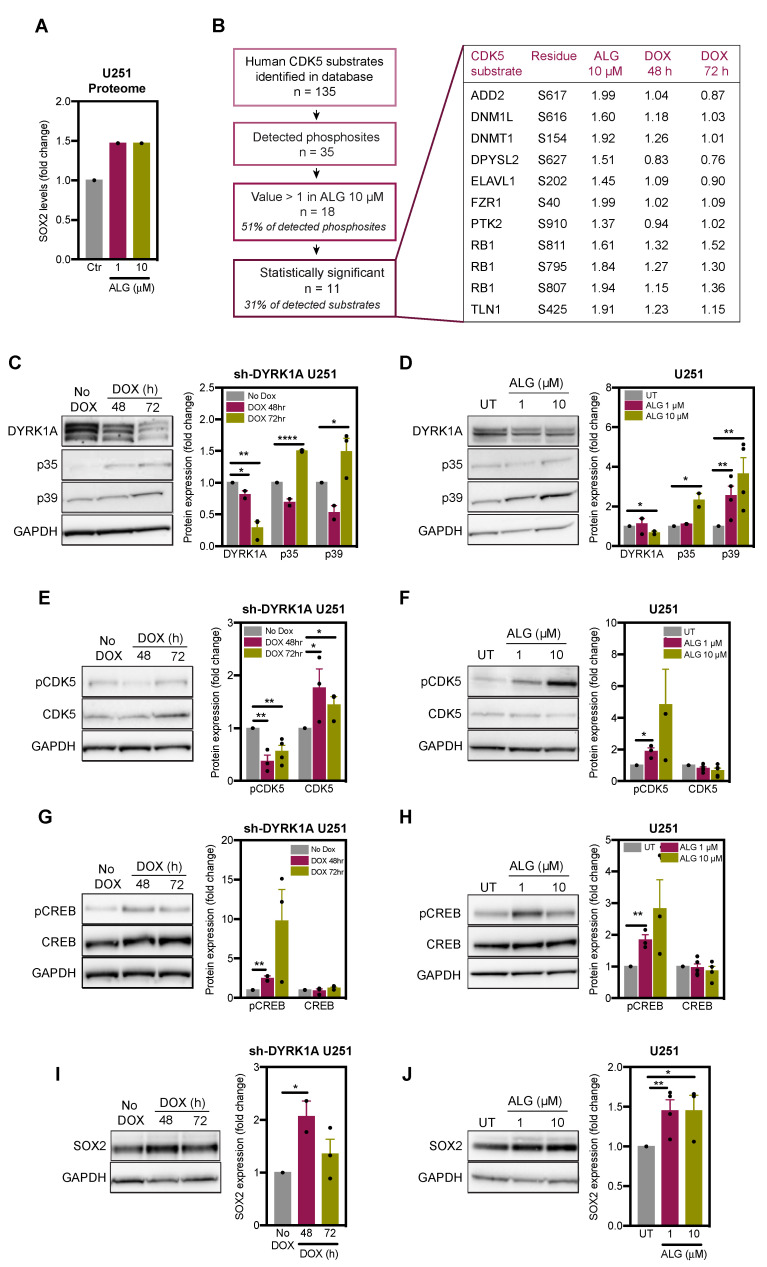
DYRK1A inhibition activates CDK5 pathway in U251 glioblastoma cells. (**A**) SOX2 expression in U251 cells treated with the DYRK1A inhibitor ALGERNON (ALG) for 72 h. Control (Ctrl) cells were left untreated. Data extracted from proteomic analysis in reference [17]. (**B**) Annotation of CDK5 substrates (derived from PhosphoSitePlus) in the phosphoproteome of U251 cells treated with ALG (72 h), and U251 cells transfected with doxycycline (DOX)-inducible shRNA targeting DYRK1A (U251 sh-DYRK1A) treated with DOX. (**C**,**D**) Immunoblot images and quantification of DYRK1A, p35 and p39 in U251 sh-DYRK1A cells treated with DOX and U251 cells treated with ALG (72 h)(*n* = 2–4). (**E**,**F**) Immunoblot images and quantification of phosphorylated (p-CDK5) and total CDK5 in U251 sh-DYRK1A cells treated with DOX and U251 cells treated with ALG (72 h)(*n* = 2–4). (**G**,**H**) Immunoblot images and quantification of phosphorylated (p-CREB) and total CREB in U251 sh-DYRK1A cells treated with DOX and U251 cells treated with ALG (72 h)(*n* = 2–4). (**I**,**J**) Immunoblot images and quantification of SOX2 in U251 sh-DYRK1A cells treated with DOX and U251 cells treated with ALG (72 h)(*n* = 2–4). All bar graphs represent mean ± SEM (two-tailed, unpaired t-test, * *p* < 0.05, ** *p* < 0.01, **** *p* < 0.0001).

**Figure 4 ijms-22-04011-f004:**
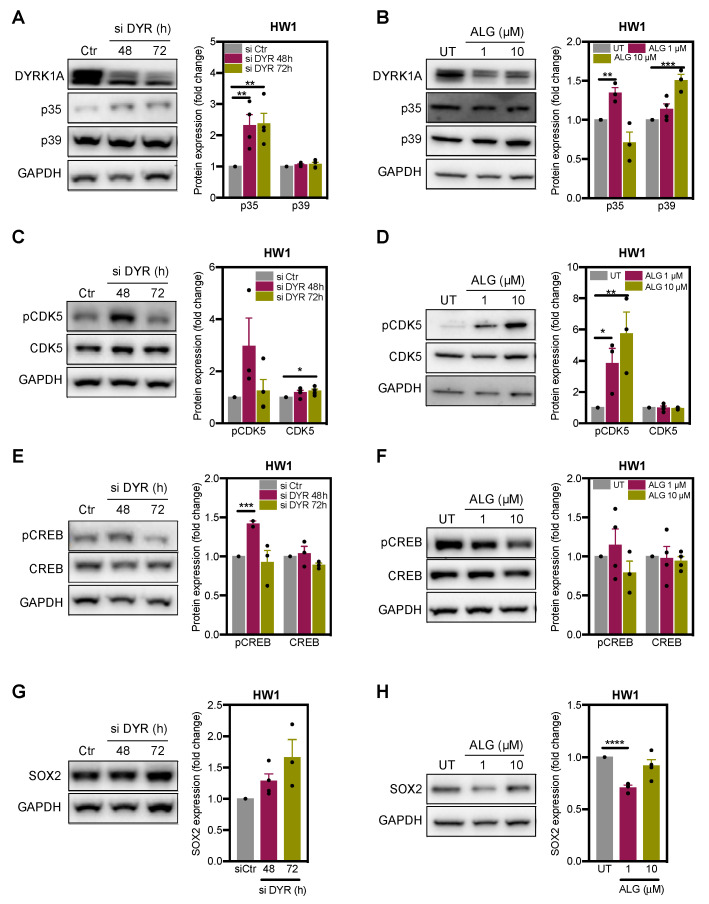
DYRK1A inhibition activates CDK5 pathway in HW1 glioblastoma stem cells. (**A**,**B**) Immunoblot images and quantification of DYRK1A, p35 and p39 in HW1 cells transfected with scramble (si Ctr) and DYRK1A (si DYR) targeting siRNAs or treated with DYRK1A inhibitor ALGERNON (ALG, 72 h)(*n* = 3–4). (**C**,**D**) Immunoblot images and quantification of phosphorylated (p-CDK5) and total CDK5 in HW1 transfected with si Ctr and si DYR, or treated with ALG (72 h)(*n* = 3–4). (**E**,**F**) Immunoblot images and quantification of phosphorylated (p-CREB) and total CREB in HW1 cells transfected with si Ctr and si DYR, or treated with ALG (72 h)(*n* = 2–4). (**G**,**H**) Immunoblot images and quantification of SOX2 in HW1 cells transfected with si Ctr and si DYR, or treated with ALG (72 h). (*n* = 3–4). All bar graphs represent mean ± SEM (two-tailed, unpaired t-test, * *p* < 0.05, ** *p* < 0.01, *** *p* < 0.001, **** *p* < 0.0001).

**Figure 5 ijms-22-04011-f005:**
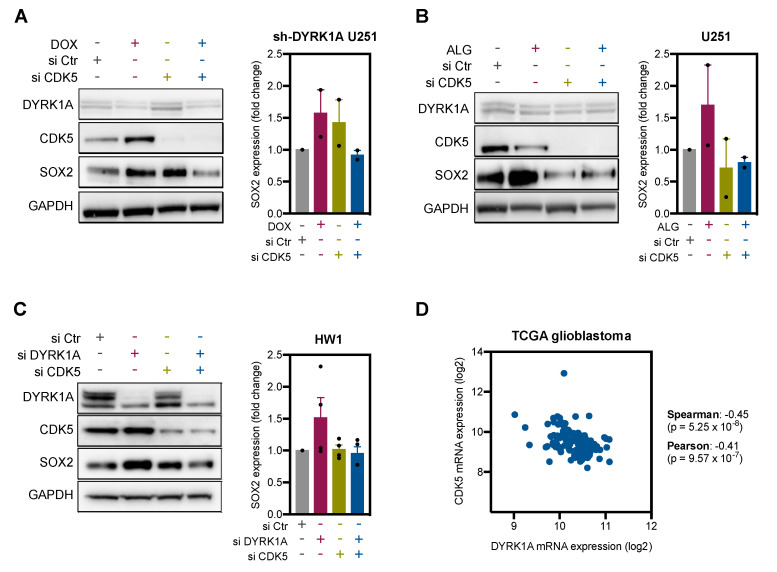
DYRK1A regulates SOX2 expression in glioblastoma cells via CDK5. (**A**) Immunoblots images of DYRK1A, CDK5 and SOX2 expression in U251 cells transfected with doxycycline (DOX)-inducible shRNA targeting DYRK1A (U251 sh-DYRK1A). CDK5-targeting (si CDK5) siRNA was employed to knock down CDK5 prior DOX-induced DYRK1A knockdown (72 h). Quantification of SOX2 is shown (*n* = 2). (**B**) Immunoblot images of DYRK1A, CDK5 and SOX2 in U251 cells transfected with si CDK5 (24 h) and treated with DYRK1A inhibitors ALGERNON (ALG, 72 h). Quantification of SOX2 is shown (*n* = 2). (**C**) Immunoblot images of DYRK1A, CDK5 and SOX2 in HW1 cells transfected with scramble (si Ctr), DYRK1A (si DYRK1A) and CDK5 (si CDK5) targeting siRNA for 72 h. Quantification of SOX2 is shown (*n* = 3–4). (**D**) Correlation of DYRK1A and CDK5 expression in the TCGA glioblastoma cohort Firehose Legacy (*n* = 136, www.cbioportal.org). All bar graphs represent mean ± SEM (two-tailed, unpaired *t*-test).

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
