# Peer review of "DYRK1A Negatively Regulates CDK5-SOX2 Pathway and Self-Renewal of Glioblastoma Stem Cells"

_ijms, 2021, doi:10.3390/ijms22084011_

Round 1

Reviewer 1 Report

The authors describe in detail experiments with cell cultures of glioblastoma stem cells with the factor DYRK1A and associated factors.

The work is a little difficult to read with numerous small pictures and diagrams, but this is due to the very careful detail work.

The results appear convincing.

Abstract and discussion are described briefly and clearly.

The results arouse interest in whether the factors described will be therapeutically applicable in the future.

Only the term “glioblastoma tumor”, which occurs twice, is unusual for neuropathologists and should be avoided (the word “glioblastoma” alone is sufficient).

Author Response

We appreciate this comment and have changed the manuscript accordingly (lines 37, 60, 73, 341).

Reviewer 2 Report

Dear Members of the Editorial Broad:

            Re: ijms-1169727 review: Entitled “DYRK1A negatively regulates CDK5-SOX2 pathway      and self-renewal of glioblastoma stem cells”.

The manuscript examines the mechanisms by which glioma stem cells (GSCs) govern their stemness and differentiation.  The authors focused on one of the kinases, DYRK1A, which has been shown to regulate the differentiation of neural progenitors, and their work does show that DYRK1A drives differentiation of GSCs and attenuates stemness of GSCs by blocking CDK5.  SOX2 is downregulated when DYRK1A inactivates CDK5, and the GSCs differentiate.

The manuscript has minor revisions:

1) Concerning Figure 1, what is the final differentiation lineage of HW1 cells?

2)  The work shows that DYRK1A controls the CDK5-CREB-SOX2 axis by reducing the expression of both p35 and p39. However, the work does not discuss or show which substrates (proteins) that DYRK1A phosphorylates to control the axis.  Based on the downstream phosphorylation consensus sequences for DYRK1A, the authors can list the putative candidates.

Author Response

1) Concerning Figure 1, what is the final differentiation lineage of HW1 cells?

Our data indicates that HW1 differentiates into a more neuronal phenotype. This comment has now been added in the manuscript (line 116).

2)  The work shows that DYRK1A controls the CDK5-CREB-SOX2 axis by reducing the expression of both p35 and p39. However, the work does not discuss or show which substrates (proteins) that DYRK1A phosphorylates to control the axis.  Based on the downstream phosphorylation consensus sequences for DYRK1A, the authors can list the putative candidates.

As indicated in our discussion (lines 300-302) we hypothesize that similar to Cyclin B, Cyclin D and p27 [13,14,17], DYRK1A regulates degradation of p35 and p39. This hypothesis is further supported by previous studies demonstrating that DYRK1A regulates the ligases RNF169 [32-34] and Von Hippel-Lindau ligase [34]. We recently discovered that DYRK1A regulates the E3-ligase Anaphase-promoting complex by phosphorylating its subunit CDC23 and identified the ligases DTX3I and TRIM56 as potential novel substrates of DYRK1A ligases [17].